# The Relationships among Chinese University EFL Learners’ Feedback-Seeking Behavior, Achievement Goals, and Mindsets

**DOI:** 10.3390/bs13020190

**Published:** 2023-02-20

**Authors:** Yunmei Sun, Yuting Huang

**Affiliations:** School of Foreign Languages, Huazhong University of Science and Technology, Wuhan 430074, China

**Keywords:** feedback-seeking behavior, achievement goals, mindsets, mediation

## Abstract

This study investigated the characteristics of feedback-seeking behavior and the underlying motivational antecedents including the mindsets and achievement goals of Chinese EFL learners. Questionnaire data were collected from 677 learners taking English classes at different levels in China for (1) their beliefs about English learning (a fixed or growth mindset), (2) goal orientation in achievement-related situations (development or demonstration goals), and (3) FSB (whether to seek feedback, by what strategies, and from whom). Results indicated that Chinese EFL learners with a growth mindset or demonstration-approach goals proactively seek feedback through variant strategies (i.e., feedback direct inquiry, indirect inquiry, and monitoring) while those with development-approach goals or a fixed mindset seek feedback by monitoring only due to learners’ different perceptions of the cost and value attached to different strategies. Furthermore, a demonstration approach partially mediated the predictive role of a growth mindset on three FSBs, while the relationships between feedback monitoring and the two mindsets were partially or fully mediated by a development approach.

## 1. Introduction

The roles of feedback and corrective feedback (CF) in performance and goal regulation over work or study have received increased attention in a wide range of disciplines in recent years [1,2]. Along with this growth of studies on feedback in the field of second language acquisition (SLA), there is also increasing concern over the passive role learners play in the process of receiving feedback about their performance from teachers. Several attempts [3,4] have been made to study the basic components and motivational antecedents (e.g., mindset, achievement goals) of foreign language learners’ feedback-seeking behavior (FSB). FSB refers to the process in which individuals will proactively monitor and seek feedback information [5]. However, no large-scale studies have been conducted to investigate the components and predictors of the FSB of Chinese EFL learners, who have been taught with the expository or duck-feeding method [6]. As a result, they have been accustomed to being passive knowledge receivers rather than active feedback seekers. This study aims to explore the relationships among Chinese university EFL learners’ feedback-seeking behavior, achievement goals, and mindsets. The results of this study might provide Chinese EFL teachers with some insights into transforming learners from passive feedback receivers into active feedback seekers.

## 2. Literature Review

### 2.1. Feedback and Corrective Feedback

A large number of studies in variant fields have proved the positive roles of feedback in performance [1] and goal regulation [2] since the early 1950s [7]. Applied to the context of language teaching, feedback, which refers to information provided to learners to improve their performance in learning tasks [8], includes assessment and correction and is considered important for L2 learning [9]. Compared to merely assessing how well or badly a learner performs, correction provides learners with more specific information to improve their language learning in ways such as an explanation or the provision of better alternatives [8], and thus corrective feedback has been a research focus in second language acquisition (SLA) for decades. CF in SLA refers to the responses to a learner’s non-target-like L2 production [10].

There are three different types of classifications for CF. From the perspective of strategies teachers applied, CF can be broadly categorized into two types: reformulations and prompts [11]. Reformulations include recasts and explicit corrections, which give learners reformulations of their non-target-like L2 production. Prompts encourage learners to self-repair and consist of variant signals such as elicitation, metalinguistic clues, clarification requests, and repetition [12]. In terms of its forms, CF can be classified into oral, written, and non-verbal CF. As regards the ideal time to give CF, there are mainly two ways to operationalize the timing of CF. The first way is to distinguish whether CF is provided while a communicative task (online feedback) is being performed or after the task is completed (offline feedback). The other way is to differentiate whether CF is offered at the early stages of instruction (immediate feedback) or after learners engage in practice activities (delayed feedback) [13].

Studies on feedback in the field of second language acquisition (SLA) have mainly focused on teachers’ providing corrective feedback to learners, and have extensively explored the conceptualization, categorization [11], antecedents [14], frequency [15], and learning outcomes [16,17] of CF, as well as related mediates of CF, including the learner’s age [12]. Previous empirical and theoretical studies conducted in laboratory or classroom settings [10] generally support the facilitating role of CF on SLA while undervaluing and even neglecting learners’ active role of seeking feedback proactively in L2 learning [3]. Therefore, Papi et al. [3] redefine corrective feedback from the perspective of social and organizational psychology and introduce the concept of feedback-seeking behavior into SLA, which casts L2 learners as human agents who can seek, attend to, and learn from different types of feedback through conscious, proactive, and selective behaviors.

### 2.2. Feedback-Seeking Behavior

#### 2.2.1. Definition, Basic Components, and Strategies of FSB

FSB refers to the process in which “individuals will actively monitor and seek feedback information concerning organizationally determined and individually held goals” [5] (p. 380). This process consists of three stages: motivation, cognitive processing, and behavior. In the first stage, individuals are considered as situated in an information environment where they proactively seek feedback due to variant motivations: competence motive, drive to self-evaluation, uncertainty reduction, and error correction (FSB) [5]. Second language learning is also considered as fundamentally a motivational pursuit and FSB as a motivated behavior [9]. The second stage involves translating those motivations into actual feedback-seeking behavior. Individuals would weigh the value against the cost related to different feedback-seeking strategies and direct their effort to decide on their feedback-seeking behavior. There are three primary costs of FSB strategies: effort cost, ego cost, and self-presentation cost [18]. Effort cost is the level of effort needed to seek feedback information [5]; ego cost relates to the risk of receiving negative feedback about oneself; and self-presentation cost refers to “the cost of exposing one’s uncertainty and need for help” [18] (p. 392). The value of feedback-seeking is concerned with the belief that seeking feedback will be useful for individuals to improve their performance and develop their abilities. In the last stage, FSB refers to the actions and strategies employed by an individual who wants to obtain information and reduce the uncertainty of his or her performance when pursuing certain goals in the field of SLA [3].

FSB consists of several basic components: the source from which or whom people seek feedback, the strategies which an individual employs to seek feedback, and the type of feedback information people seek [19]. Firstly, people can seek feedback from supervisors, coworkers, customers, or documentation like manuals and memos. SLA researchers found that L2 learners tended to seek feedback from teachers and others (e.g., peers, friends, and native speakers of their target languages) [3], including socializing agents such as senior researchers and former colleagues in the same field, a university’s writing support services, and a native speaker’s editing services from a private company [20].

Secondly, Ashford and Cummings [5] offered two sets of strategies individuals used to seek feedback: feedback monitoring and feedback inquiry. The former implies that individuals actively attend to and take information from the environment by observing the situation and other actors’ behaviors based on their self- and goal-related schemas, while the latter is associated with “asking actors in that environment for their perception and/or evaluation of the behavior in question” [5] (p. 385). Miller and Jablin [21] argued that feedback inquiry could be further divided into asking for feedback directly or indirectly (e.g., by using hinting, joking, or roundabout questions). Learners from different cultures display a distinctive preference for strategies used to receive and seek feedback. For example, learners from collective cultures (e.g., Confucian-based Asia) prefer to receive indirect and implicit feedback, while those from individualist cultures (e.g., the United States) are more likely to employ a direct inquiry strategy to seek feedback [22].

Thirdly, the types of feedback information individuals seek could range from performance feedback, referent feedback, technical feedback, and social feedback, to normative feedback [19]. However, L2 learners mainly seek performance feedback, strategic feedback, and corrective feedback. More specifically, they actively seek performance feedback from variant sources to evaluate their learning performance [23] and strategic feedback information to determine the appropriateness of the learning methods adopted to learn a foreign or a second language [19]; they consciously seek and learn from corrective feedback, (responses to a learner’s non-target like L2 production) from people they trust to improve their L2 proficiency [3].

#### 2.2.2. Previous Studies on FSB

Since the concept of FSB was proposed, research on FSB has covered dimensions such as conceptualization, measurement [23,24], motivational mechanisms [25,26], individual and situational antecedents, and outcomes of FSB. Studies on FSB have been conducted in fields like management [25], psychology [27], and education [28], all of which have demonstrated the value of FSB for both individuals and organizations [19].

While in the field of SLA, it was not until 2019 that Papi et al. [3] first investigated the basic components and motivational antecedents of L2 learners’ FSB. Ever since then, the foci of the limited number of FSB studies in SLA include the conceptualization, classification, and antecedents of feedback-seeking strategies. The antecedents of FSB that have been examined include individual efforts, affective differences (e.g., self-efficacy) [29], conative differences (e.g., mindsets and achievement goals) [3,4], available resources, and contextual and structural factors [20]. For example, Gan et al. [29] probed into individual differences in learners’ choices of seeking, avoiding [30], and acting on teacher feedback, while Sung [20] focused on learners’ sources of feedback. Even so, there is still a very limited number of studies on FSB, and the existing research either focused on English speakers’ FSB in the SLA process [3,4] or had a very limited number of participants, leaving the findings hard to generalize [20]. All neglect the learners’ feedback-seeking strategies, which might be of significant pedagogical implications for those such as Chinese EFL learners who have been instructed under a test-oriented educational system and are accustomed to being passive receivers of feedback.

### 2.3. Achievement Goals

#### 2.3.1. Definition and Classification of Achievement Goals

Achievement goals are concerned with personal goal preference in achievement-related situations [31] and are traditionally divided into learning goals and performance goals. Learning goals are associated with increasing competence through acquiring new skills and showing mastery-oriented response patterns including seeking challenging tasks [32]. Performance goals, however, are related to demonstrating as well as validating the adequacy of an individual’s competence by seeking favorable judgments and avoiding negative judgments on one’s competence [31]. Korn and Elliot [33] added a dichotomy to the traditional classification of achievement goals, which bifurcate learning and performance goals (relabeled as development and demonstration goals, respectively) into two types concerning their valence (i.e., approach vs. avoidance). Development-approach and development-avoidance goals are concerned with improving and maintaining competence, respectively [33]. Performance-approach and performance-avoidance refer to the demonstration of competence and the avoidance of demonstrating incompetence compared to other learners, respectively [34].

#### 2.3.2. Previous Studies on FSB and Achievement Goals

Research attention on FSB began with a focus on its individual (also called seeker), target (source), contextual antecedents, and outcome variables [35]. VandeWalle [36] studied individual variables and maintained that achievement goals are important motivational constructs underlying the quality and quantity of one’s FSB. Research on the relationship between FSB and achievement goals started by establishing an overall construct of FSB and confirmed a positive correlation between the frequency of FSB and the achievement of learning goals [18]. Later on, researchers noticed the divergences among different feedback-seeking strategies and attempted to probe into the outcomes and antecedents of specific feedback-seeking strategies. For example, Papi et al. [3] examined the predictive role of achievement goals on FSB and concluded that L2 learners with development goals tend to seek feedback through both inquiry and monitoring strategies from teachers as well as others, while learners with demonstration goals only seek feedback through inquiry.

### 2.4. Mindsets

#### 2.4.1. Definition and Classification of Mindsets

Mindsets refer to people’s beliefs about the malleability of their intelligence, that is, whether individuals believe their intelligence can develop and improve through effort and practice or not [32]. Mindsets were traditionally divided into growth and fixed mindsets [37]. A growth mindset (or incremental theory of intelligence) is defined as a belief that construes intelligence as malleable and improvable [38]. Learners with a growth mindset are likely to learn through a mastery approach, embracing challenges and putting in the effort to learn [39]. In comparison, learners with a fixed mindset (also called an entity theory of intelligence) believe that one’s basic qualities and attributes such as aptitude and intelligence are innate and, therefore, unchangeable [40]. A fixed mindset was related to “adopting the performance goal of documenting the entity”, whereas a growth mindset was attached to “the learning goal of developing that quality” [32] (p. 256).

#### 2.4.2. Previous Studies among Mindsets, Achievement Goals, and FSBs

Mindset theory, concerning one’s fundamental beliefs about the malleability of personal characteristics, has been applied to variant domains such as psychology, relationships, and management for a long time. However, SLA researchers did not systematically examine the role of mindsets in language learning until the past decade [40]. To date, studies on mindsets in the SLA field have mainly focused on whether and how individual differences in mindsets are related to human motivation, including distinct cognitions, emotions, and behaviors, as well as ultimate achievement and resilience [40]. For example, a growth mindset was found to be positively associated with effort belief, learning goals, and mastery-oriented strategies like feedback-seeking, while it was negatively associated with setting goals that focused on failure avoidance, avoidant coping strategies, and anxiety [40]. Other researchers such as Papi et al. [4] found that a growth mindset predicted L2 learners’ perception of the value of feedback, which positively predicted both feedback inquiry and feedback monitoring, while learners’ fixed mindset predicted the cost of feedback, which is a negative predictor of feedback monitoring.

Dweck and Leggett [32] maintained that learners’ adoption of different achievement goals could have been rooted in their mindset. Studies by Papi et al. [3] also confirmed the mediating role of achievement goals on the relationship between mindsets and FSB. They found that development-approach goals partially mediated the relationship between the growth mindset and three feedback-seeking strategies (i.e., feedback monitoring, feedback inquiry/teacher, feedback inquiry/others), indicating that learners with a growth mindset and development-approach goals tend to seek feedback that is conducive to the development of their competence through different strategies and from variant sources.

Taken together, previous studies demonstrate that there exists a relationship between FSB and two motivational antecedents: mindsets and achievement goals. However, further empirical research is needed to verify the preliminary findings. The above review of the literature also revealed a lack of research on EFL learners from mainland China. Furthermore, it is necessary to investigate the nature and role of Chinese EFL learners’ FSB for at least two reasons. First, most Chinese EFL learners have been accustomed to the expository or duck-feeding method of teaching [6], in which they are passive knowledge receivers rather than active feedback seekers. Second, Chinese EFL learners are considered to have the fear of losing face (also called *lian*, i.e., fear of being laughed at and feeling embarrassed), which restrains learners from seeking feedback [41] in the second/foreign learning process [42]. The educational and social differences may contribute to Chinese EFL learners’ perception of a higher face-loss cost related to seeking feedback through direct inquiry [19]. The results of this study could help enhance understanding as well as raise the attention of learners, instructors, researchers, and policymakers to the nature and motivational mechanism of FSB in the English-learning process of Chinese learners. In light of the need for more research into Chinese EFL learners’ FSB and its relationship with the two motivational antecedents, this study addresses the following research questions:

Research Question 1: What are the relationships between Chinese EFL learners’ achievement goals and FSB?

Research Question 2: What are the relationships between Chinese EFL learners’ mindsets and FSB?

Research Question 3: Are the possible relationships between Chinese EFL learners’ mindsets and FSB mediated by their achievement goals?

## 3. Methodology

### 3.1. Participants

Six hundred and ninety Chinese university students from different parts of China, including southern, eastern, and central regions, were recruited for this study. All of them have studied English for at least 6 years. After eliminating data from 13 respondents because of random answering, the final samples included 677 participants, with 287 males and 390 females. There were 566 undergraduates (431 freshmen, 58 sophomores, and 77 others), 107 graduates, and 4 doctoral students from different disciplines, covering engineering, management, and linguistics. They are all Chinese native speakers. Only several of them had the experience of living or studying in an English community for a few months.

### 3.2. Instrumentation

To address the three research questions, a questionnaire consisting of four parts was adapted and applied to measure Chinese EFL learners’ mindsets, achievement goals, and FSBs, respectively, and some basic demographic information such as gender, age, etc., is also included. The instruments used were adapted mainly from Papi et al.’s 2019 version, taking Chinese EFL learners’ context and Krasman’s 2010 model into consideration.

The items of mindset and achievement goals were responded to using a 6-point Likert scale, with 1 indicating strongly disagree, 2 indicating disagree, 3 indicating mostly disagree, 4 indicating mostly agree, 5 indicating agree, and 6 indicating strongly agree. The scale of mindsets included 8 items directly adapted from Papi et al.’s [3], changing language learning intelligence into English learning intelligence. The adapted scale of achievement goals was composed of 12 items, aiming to measure English-specific achievement goals. The items of FSB were responded to using a 5-point Likert scale to investigate the frequency of learners’ FSB, with 1 indicating never, 2 indicating seldom, 3 indicating sometimes, 4 indicating often, and 5 indicating always. All three original sub-scales were written in English, so we first translated each item from English into Chinese to avoid any possible misunderstandings caused by language competence and then re-translated them back into English to verify the reliability of the scales. We also invited experts in the field of translation study to proofread our translated work. The reliability of the questionnaire was high (i.e., Cronbach’s alpha > 0.90) or good (i.e., Cronbach’s alpha > 0.80) because the Cronbach’s alpha coefficients were 0.90, 0.93, 0.81, and 0.90, respectively, for the whole scale and the sub-scales of FSBs, mindsets, and achievement goals.

Aiming at figuring out the latent variables underlying the scales of Chinese EFL learners’ FSB, mindsets, and achievement goals, we conducted two exploratory factor analyses (EFA) for all three sub-scales for data reduction using SPSS 21 (IBM). In the first EFA, the method of maximum likelihood was used for extraction, direct oblimin with Kaiser normalization was used for rotation, and eigenvalues larger than 1 (Kaiser’s criterion), and scree plots were used to determine the number of factors. Results indicated that the numbers of factors presented by eigenvalues larger than 1 and the scree plots were contradictory. Based on the theoretical framework confirmed in previous studies [3,33], we conducted the second EFA with a predetermined number of factors’ extraction. Before the formal analyses, five items on FSB were deleted because of repeatability and cross-loadings. EFAs conducted on the instrument of FSB yielded three factors that explained 60.7% of the variance. The first factor, composed of six items on learners’ tendency to seek feedback indirectly (e.g., by hinting, joking, or asking roundabout questions) from teachers or others, was named Feedback Indirect Inquiry. The second factor, labeled as Feedback Monitoring, consists of five items on learners’ conscious employment of attentional resources to monitor the feedback present in the environment [3,23]. The third factor included eight items on learners’ direct inquiry for feedback from teachers and others and was labeled as Feedback Direct Inquiry. The Cronbach’s alpha coefficients of all three factors reached 0.90, 0.86, and 0.90, respectively.

The EFAs for the instrument of achievement goals yielded four factors explaining 64.9% of the variance. The four factors were labeled as Development-Approach, Demonstration-Avoidance, Development-Avoidance, and Demonstration-Approach, demonstrating learners’ valence (approach vs. avoidance) towards different (demonstration vs. development) achievement goals [33]. The Cronbach’s alpha coefficients reached 0.83, 0.82, 0.86, and 0.75, respectively, for each factor of the scale, indicating good or acceptable internal consistency of this scale. EFAs conducted on the instrument of mindsets yielded two factors which were named Growth Mindset and Fixed Mindset, explaining 56.7% of the variance. The Cronbach’s alpha coefficients reached 0.85 and 0.81 for each factor of the mindset scale, respectively. Each factor of achievement goals and mindsets includes three and four items, respectively. For details, see Appendix A, Appendix B and Appendix C.

### 3.3. Data Collection and Analysis

The adapted questionnaire was distributed to the 690 recruited participants online in the spring of 2021. Participants were informed of the purpose and their rights of voluntary participation and confidentiality on the first page of the questionnaire. Before formal analysis, we examined the raw data and eliminated 13 random answers (selecting the same answer for all the items), leaving 677 final samples.

To address the first and second research questions, we conducted several multiple regression analyses using the stepwise method, which combines the advantages of forward and backward regression [43]. We set achievement goals and mindsets as antecedents and FSBs as outcome variables. The Bonferroni adjustment was applied to avoid Type I errors resulting from multiple significance testing [3]. Descriptive statistical analyses, assumptions of normality, correlation, and collinearity among variables were tested before or with the regression analyses.

Achievement goals are considered not only as motivational antecedents of FSB [36] but also as possible outcome variables of mindsets [32].

To answer the third research question, mediation models proposed by Baron and Kenny [44] are used to analyze the mediating role of achievement goals. Three conditions have to be met. That is, a variable may function as a mediator when (a) the independent variable (i.e., mindsets) significantly predicts the presumed mediator (i.e., achievement goals), (b) the presumed mediator significantly predicts the dependent variable (i.e., FSB), and (c) the previously significant relation between the independent variable and the dependent variable eliminates or decreases if the presumed mediator’s effect on the dependent variable is controlled. More specifically, full mediation holds when the previously significant predictive impact of the independent variable on the dependent variable is no longer significant after controlling the impact of the mediator, and partial mediation holds when the effect of the independent variable on the dependent variable remains significant albeit reduced if the mediator’s impact is controlled [3]. The Sobel test of mediation was employed to examine the hypothesized mediations (i.e., whether a drop in the beta value is significant or not) avoiding subjective judgment of partial mediation.

## 4. Results

### 4.1. Descriptive Statistics, Normality Tests, and Correlations

Table 1 displays the descriptive statistics. The mode of different variables indicated that most Chinese EFL learners considered themselves as having a growth mindset and four achievement goals. In addition, the mean and standard deviation (SD) of the dataset indicated that Chinese EFL learners generally tended to possess a growth mindset, pursued development goals, and sought feedback through monitoring. A normal distribution of the datasets was indicated by the coefficients of skewness and kurtosis (absolute value < 1).

Table 2 shows that most factors of each variable were significantly correlated to each other (r < 0.7, *p* < 0.05), ranging from low (r ≤ 0.39) to moderate correlation (0.40 ≤ r ≤ 0.69), which laid the foundation for the following regression analyses. In the field of SLA, we commonly accept correlations between variables that are not very high because they might affect the process of foreign language learning, and we cannot ignore the important role of these variables even with not-very-high correlations [43].

### 4.2. Predictive Roles of Chinese EFL Learners’ Achievement Goals on FSB

The coefficients of tolerance and VIF in the following tables indicate no collinearity. The stepwise regression with Bonferroni adjustment in Table 3 shows that Demonstration-Approach predicted three FSBs significantly (*p* < 0.05), while Development-Approach only significantly predicted Feedback Monitoring (*p* < 0.001).

### 4.3. Predictive Roles of Chinese EFL Learners’ English Mindsets on FSB

Table 4 shows that Growth Mindset significantly predicted three FSBs (*p* < 0.001), while Fixed Mindset emerged as a significant predictor of Feedback Monitoring only (*p* < 0.05).

### 4.4. Mediating Roles of Achievement Goals on Relationships between Chinese EFL Learners’ Mindsets and FSB

To examine mediation models, several stepwise regression analyses were then conducted to figure out variables that can meet both condition (b) (see Table 3) and condition (a) of the above-mentioned presumed mediators [44], that is, to test the predictive roles of Chinese EFL learners’ mindsets on their achievement goals. The results in Table 5 indicated that Growth Mindset significantly (*p* < 0.001) predicted four achievement goals, while Fixed Mindset only significantly predicted Development-Approach (*p* < 0.05) and Demonstration-Avoidance (*p* < 0.001).

After figuring out all the variables that met both condition (a) and condition (b), four stepwise regression analyses were conducted to examine the mediating role of Demonstration-Approach on the relationship between Growth Mindset and three FSBs, and whether Development-Approach mediates the predictive roles of two mindset scales on Feedback Monitoring.

Table 6 and Figure 1 show that Demonstration-Approach partially mediated the predictive role of Growth Mindset on Feedback Direct Inquiry (Sobel statistic = 5.28, *p* < 0.001), Indirect Inquiry (Sobel statistic = 5.35, *p* < 0.001), and Feedback Monitoring (Sobel statistic = −2.52, *p* < 0.05), with a significant drop of the beta value of Growth Mindset for outcome variables. Growth Mindset and Demonstration-Approach together explained 7% of the variance in Feedback Direct Inquiry (R² = 0.07, F = 26.364, *p* < 0.001), explained 8% (R² = 0.08, F = 28.862, *p* < 0.001) of the variance in Feedback Indirect Inquiry, and accounted for 5% (R² = 0.05, F = 20.007, *p* < 0.001) of the variance in Feedback Monitoring.

Development-Approach emerged as a full mediator between Growth Mindset and Feedback Monitoring, making the former no longer a significant predictor of the latter. Development-Approach also partially mediated the predicting role of Fixed Mindset on Feedback Monitoring, with a significant decrease in the beta weight (Sobel statistic = −2.67, *p* < 0.05). Development-Approach and Fixed Mindset together explained 22% of the variance in Feedback Monitoring (R² = 0.22, F = 99.032, *p* < 0.001).

## 5. Discussion

### 5.1. The Relationships between Chinese EFL Learners’ Achievement Goals and FSB

The first research question addressed the relationship between Chinese EFL learners’ achievement goals (Development-Approach, Demonstration-Avoidance, Development-Avoidance, and Demonstration-Approach) and FSB (Feedback Monitoring, Feedback Direct Inquiry, and Feedback Indirect Inquiry). The results showed that learners’ approach goals significantly predicted their FSBs. More specifically, Demonstration-Approach positively predicted both Direct and Indirect Feedback Inquiry but negatively predicted Feedback Monitoring. However, Development-Approach was found only to positively predict Feedback Monitoring, which was different from previous research findings [3].

One plausible explanation is the influence of the distinctive and conscious cost-value calculations associated with different feedback-seeking behaviors of Chinese EFL learners who possess variant achievement goals [3]. In this regard, Chinese EFL learners who have been accustomed to examination-oriented and expository education [6] may perceive higher ego, effort, and self-presentation costs (which refer to the risk of receiving negative feedback about oneself, the level of effort needed to seek feedback information, and the cost of disclosing one’s uncertainty and need for help, respectively) [5,18] for feedback than learners from other cultures [3,4]. On one hand, learners with development-approach goals who simply want to develop their abilities may avoid seeking feedback through inquiries because they attach a higher face-loss cost of feedback to inquiry than monitoring strategies. Thus, they tend to use feedback monitoring to observe the situation and the behaviors of teachers or other language learners because it is a high-value and low-cost strategy [36]. On the other hand, learners who want to demonstrate their competence are more likely to seek feedback information through direct and indirect inquiry. This means learners may consider FSB as an interactive tool to make a good and positive impression on teachers and their peers through inquiries. In other words, learners with demonstration-approach goals perceive higher impression-management value over the ego, self-presentation, and effort cost of FSB through direct or indirect inquiry [3]. However, they may avoid applying feedback monitoring because simple monitoring does not offer learners interactive opportunities to impress both teachers and their peers, Additionally, the fact that Development-Avoidance and Demonstration-Avoidance did not predict Chinese EFL learners’ FSB suggests that learners perceive a higher cost than the value of FSB when they want to maintain existing competence or demonstrate that they are not incompetent compared to other learners. Our results confirmed that learners’ achievement goals play an important role in activating learners’ FSB, which is a proactive and self-motivated behavior requiring learners’ strong motivation to improve their ability or competence.

### 5.2. The Relationships between Chinese EFL Learners’ Mindsets and FSB

The second research question addressed the relationship between Chinese EFL learners’ mindsets (fixed vs. growth mindset) and FSB. The results revealed that Growth Mindset positively predicted Chinese EFL learners’ three FSBs significantly, while Fixed Mindset negatively predicted Feedback Monitoring. The findings are consistent with those of Papi et al. [3,4]. Our findings further confirm that learners with a growth mindset perceive a higher value over the cost of feedback, and they tend to proactively seek feedback from variant resources (e.g., teachers, peers, native speakers, or other more proficient English users) through different strategies (i.e., direct inquiry, indirect inquiry, and monitoring). The results also demonstrate that learners with a fixed mindset consider their intelligence and ability to learn English as unchangeable. Accordingly, they think their effort to seek feedback is unavailing.

### 5.3. Mediating Role of Achievement Goals on the Relationships between Chinese EFL Learners’ Mindsets and FSB

The third research question addressed whether achievement goals mediated the relationships between learners’ mindsets and FSB. Results indicated that Demonstration-Approach partially mediated the predictive role of Growth Mindset on three FSB factors, but Development-Approach fully mediated the relationship between Growth Mindset and Feedback Monitoring. These results contradict Papi et al.’s [3] findings which indicated the mediative roles of development-approach goals and two demonstration goals on the relationships between mindsets and FSB. Our results further confirm that learners with a growth mindset can have both development-approach and demonstration-approach goals [3]. However, our results contradict their results in finding that demonstration goals should not be the main occupation of learners. Compared to second language learners in the US, Chinese EFL learners with a growth mindset and demonstration-approach goals are more likely to seek feedback through variant strategies, while those with development-approach goals tend to seek feedback only through monitoring.

A possible explanation for the differences might be that Chinese EFL learners with a growth mindset may perceive a similar level of competence development, uncertainty reduction, and error correction value of FSB. However, their distinctive achievement goals may contribute to a different perception of the value and cost attached to variant feedback-seeking strategies. For instance, learners chasing demonstration-approach goals attach higher performance and impression-management value to feedback inquiries than those with development-approach goals. On the contrary, learners with development-approach goals are less likely to use feedback-seeking inquiry due to its higher cost compared to monitoring. The full mediating role of Development-Approach on the relationship between Growth Mindset and Feedback Monitoring indicated that it is a strong indicator of a mediative effect [44].

The results also demonstrated that Development-Approach partially mediated the relationship between Fixed Mindset and Feedback Monitoring. These results contradict previous findings [3,4]. One plausible explanation could be that learners with a fixed mindset think their ability to learn a foreign language is unmalleable and thus perceive FSB as a low-value activity. In addition, previous studies proved that Fixed Mindset positively predicted the self-presentation cost of FSB, which negatively predicted Feedback Monitoring [4]. However, Chinese EFL learners were situated in a test-oriented educational system, in which individuals need to try their best to improve their competence even though they are not confident enough about their English competence through their effort. The pressure of taking and passing numerous tests together with peer pressure urged them to actively seek feedback from people around them. Considering the low value of FSB and high costs of inquiries, they tend to monitor the environment and other actors’ (i.e., teachers or more advanced English learners) reactions towards their or their peers’ performance to obtain feedback information.

## 6. Educational Implications

The results further proved the important role of mindsets and achievement goals in foreign language learners’ FSB as well as the mediative role of Development-Approach in the relationships between two mindset factors and Feedback Monitoring strategies. Therefore, our findings could have some pedagogical implications for foreign language educators.

First, foreign language educators could intervene by prompting learners’ growth mindset and development-approach goals. This might increase the value and decrease the cost of FSB, thus contributing to learners’ FSB. For instance, teachers could set both short-term and long-term goals for learners and focus on developing their learning ability to achieve these goals rather than set performance standards. This could help minimize the degree of competition and social comparison, thus lessening the pressure and enhancing learners’ confidence.

Second, teachers could also create an FSB-friendly environment by developing a favorable seeker–source relationship, thus stimulating learners’ motivation to seek feedback from different sources using variant strategies. For example, they could seek help from peers, friends, and teachers in private rather than in class or public.

Third, teachers could help Chinese EFL learners with development-approach goals but a fixed mindset to attribute the progress of their English performance to their effort. For example, teachers could encourage learners to set development-approach goals and record their effort and progress, which may help learners realize the malleability of their foreign language intelligence through effort, hence forming a growth mindset. This may help them find it more useful and meaningful to proactively seek feedback from the bottom of their heart.

## 7. Conclusions

This study investigated the predictive role of achievement goals and mindsets on Chinese EFL learners’ FSB. We found learners with different achievement goals and mindsets tend to make conscious calculations of the cost and value attached to variant feedback-seeking strategies. More specifically, Demonstration-Approach and Growth Mindset positively predicted learners’ feedback-inquiry strategies, while Feedback Monitoring was positively predicted by Development-Approach and Growth Mindset and negatively predicted by Demonstration-Approach and Fixed Mindset. We also found the relationships between Growth Mindset and three FSB factors were mediated by Demonstration-Approach, while Development-Approach fully or partially mediated the predictive role of two mindset factors on Feedback Monitoring.

### Limitations and Directions for Future Research

The present study only employed a self-report questionnaire to conduct quantitative analyses of the relationships among Chinese EFL learners’ FSB, mindsets, and achievement goals. Therefore, our future study would employ a mixed method combining self-reported questionnaire data with qualitative data such as interviews or learners’ diaries, which might yield richer findings. Furthermore, future research could also investigate the mediating roles of FSB on other variables such as motivation, language aptitude, proficiency levels, the teacher–student relationship, etc., as well as interventions that might improve L2 learners’ FSB in their language learning process.

## Figures and Tables

**Figure 1 behavsci-13-00190-f001:**
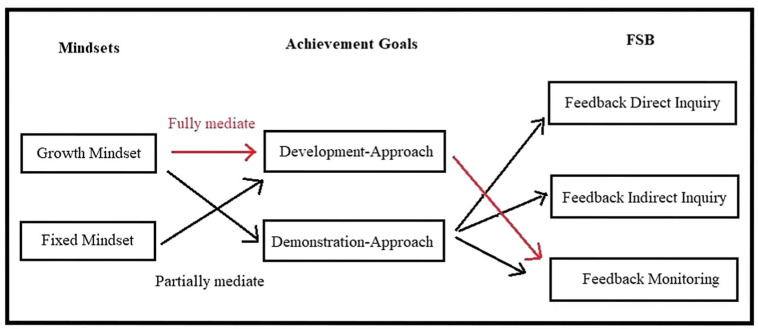
Mediating Role of Achievement Goals on Relationships Between Mindset and FSB.

**Table 1 behavsci-13-00190-t001:** Descriptive Statistics of Learners’ Mindsets, Achievement Goals, and FSBs.

	Min.	Max.	Mode	Mean	SD	Skewness (Std. Error)	Kurtosis (Std. Error)
Growth Mindset	1	6	4	3.77	0.97	−0.01 (0.09)	0.38 (0.19)
Fixed Mindset	1	6	3	3.06	0.99	−0.08 (0.09)	−0.11 (0.19)
Development-Approach	1	6	4	4.51	0.84	−0.31 (0.09)	0.70 (0.19)
Demonstration-Avoidance	1	6	4	3.88	1.02	−0.12 (0.09)	0.19 (0.19)
Development-Avoidance	1	6	4	4.22	0.90	−0.15 (0.09)	0.41 (0.19)
Demonstration-Approach	1	6	4	3.75	0.94	−0.07 (0.09)	0.56 (0.19)
Feedback Direct Inquiry	1	5	3	2.45	0.82	0.36 (0.09)	−0.13 (0.19)
Feedback Indirect Inquiry	1	5	3	2.31	0.89	0.45 (0.09)	−0.26 (0.19)
Feedback Monitoring	1	5	3	3.39	0.84	−0.38 (0.09)	0.08 (0.19)

**Table 2 behavsci-13-00190-t002:** Intercorrelations Among Predictors and Outcome Variables.

	1	2	3	4	5	6	7	8
1. Growth Mindset	1							
2. Fixed Mindset	−0.31 **	1						
3. Development-Approach	0.42 **	−0.21 **	1					
4. Demonstration-Avoidance	0.15 **	0.10 **	0.36 **	1				
5. Development-Avoidance	0.32 **	−0.08 *	0.65 **	0.59 **	1			
6. Demonstration-Approach	0.27 **	0.02	0.44 **	0.65 **	0.52 **	1		
7. Feedback Direct Inquiry	0.19 **	−0.04	0.16 **	0.12 **	0.13 **	0.24 **	1	
8. Feedback Indirect Inquiry	0.16 **	0.03	0.05	0.18 **	0.12 **	0.27 **	0.75 **	1
9. Feedback Monitoring	0.22 **	−0.19 **	0.47 **	0.12 **	0.29 **	0.14 **	0.41 **	0.30 **

**. Correlation is significant at the 0.01 level (2-tailed). *. Correlation is significant at the 0.05 level (2-tailed).

**Table 3 behavsci-13-00190-t003:** Predictive Roles of Chinese EFL Learners’ Achievement Goals on FSB.

Outcome Variable	R^2^	Predictor	B	Std. Error	*β*	t	Sig.	Collinearity Statistics	
								Tolerance	VIF
Feedback Direct Inquiry	0.06	(Constant)	1.67	0.13		13.34	0.001		
		Demonstration-Approach	0.21	0.03	0.24	6.40	0.001	1.00	1.00
Feedback Indirect Inquiry	0.07	(Constant)	1.37	0.14		1.12	0.001		
		Demonstration-Approach	0.25	0.03	0.27	7.15	0.001	1.00	1.00
Feedback Monitoring	0.22	(Constant)	1.40	0.16		8.51	0.001		
		Development-Approach	0.51	0.04	0.51	13.41	0.001	0.80	1.25
		Demonstration-Approach	−0.08	0.03	−0.09	−2.38	0.018	0.80	1.25

**Table 4 behavsci-13-00190-t004:** Predictive Roles of Chinese EFL Learners’ English Mindsets on FSB.

Outcome Variable	R^2^	Predictor	B	Std. Error	*β*	t	Sig.	Collinearity Statistics	
								Tolerance	VIF
Feedback Direct Inquiry	0.03	(Constant)	1.86	0.12		14.92	0.001		
		Growth Mindset	0.16	0.03	0.18	4.88	0.001	1.00	1.00
Feedback Indirect Inquiry	0.02	(Constant)	1.75	0.14		12.84	0.001		
		Growth Mindset	0.15	0.04	0.16	4.24	0.001	1.00	1.00
Feedback Monitoring	0.06	(Constant)	3.12	0.19		16.46	0.001		
		Growth Mindset	0.16	0.03	0.18	4.69	0.001	0.90	1.11
		Fixed Mindset	−0.11	0.03	−0.13	−3.30	0.001	0.90	1.11

**Table 5 behavsci-13-00190-t005:** Predictive Role of Chinese EFL Learners’ Mindsets on Achievement Goals.

Outcome Variable	R^2^	Predictor	B	Std. Error	*β*	t	Sig.	Collinearity Statistics	
								Tolerance	VIF
Development-Approach	0.18	(Constant)	3.45	0.18		19.48	0.001		
		Growth Mindset	0.34	0.03	0.39	1.62	0.001	0.90	1.11
		Fixed Mindset	−0.07	0.03	−0.08	−2.30	0.022	0.90	1.11
Demonstration-Avoidance	0.04	(Constant)	2.59	0.23		11.13	0.001		
		Growth Mindset	0.21	0.04	0.20	4.95	0.001	0.90	1.11
		Fixed Mindset	0.17	0.04	0.16	4.12	0.001	0.90	1.11
Development-Avoidance	0.10	(Constant)	3.11	0.13		23.49	0.001		
		Growth Mindset	0.30	0.03	0.32	8.67	0.001	1.00	1.00
Demonstration-Approach	0.07	(Constant)	2.74	0.14		19.48	0.001		
		Growth Mindset	0.27	0.04	0.27	7.39	0.001	1.00	1.00

**Table 6 behavsci-13-00190-t006:** Mediating Role of Achievement Goals on Relationships Between Mindset and FSB.

Outcome Variable	R^2^	Predictor	B	Std. Error	*β*	t	Sig.	Collinearity Statistics	
								Tolerance	VIF
Feedback Direct Inquiry	0.07	(Constant)	1.38	0.15		9.00	0.001		
		Demonstration-Approach	0.18	0.03	0.20	5.29	0.001	0.93	1.08
		Growth Mindset	0.11	0.03	0.13	3.34	0.001	0.93	1.08
Feedback Indirect Inquiry	0.08	(Constant)	1.13	0.17		6.82	0.001		
		Demonstration-Approach	0.23	0.04	0.24	6.23	0.001	0.93	1.08
		Growth Mindset	0.09	0.04	0.10	2.48	0.013	0.93	1.08
Feedback Monitoring	0.22	(Constant)	1.61	0.20		8.13	0.001		
		Development-Approach	0.45	0.03	0.45	12.95	0.001	0.96	1.04
		Fixed Mindset	−0.08	0.03	−0.09	−2.73	0.007	0.96	1.04
	0.07	(Constant)	2.93	0.20		14.32	0.001		
		Growth Mindset	0.13	0.04	0.15	3.76	0.001	0.83	1.21
		Fixed Mindset	−0.12	0.03	−0.14	−3.59	0.001	0.89	1.12
		Demonstration-Approach	0.09	0.04	0.10	2.49	0.013	0.91	1.10

## Data Availability

The data presented in this study are available on request from the corresponding author. The data are not publicly available because we still need the data for further research.

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
