# Peer review of "The Relationships among Chinese University EFL Learners’ Feedback-Seeking Behavior, Achievement Goals, and Mindsets"

_behavsci, 2023, doi:10.3390/bs13020190_

Round 1
Reviewer 1 Report
This study investigated the relationships between feedback-seeking behavior, mindsets and achievement goals among Chinese EFL learners. As studies of these relationships have rarely been conducted in China, the study makes a contribution to the field. Specific comments are as follows:
In the section of instruments, you may explain how you adapted the items of of mindset and achievement goals, as well as items of FSB from Papi et (2019).
Statistics about Mediating Role of Achievement Goals on Relationships Between Mindset and FSB were presented in Table 6. If possible, you may present these results in Figures which may help readers understand the results more easily.
In the Discussion part, on p.10, you wrote : "Chinese EFL learners who have been accustomed to examination-oriented and expository education (Liu, 2010) 391 may perceive higher ego, effort, and self-presentation cost of feedback than learners from other cultures". Perhaps it pays to mention concepts like "ego, effort, and self-presentation cost of feedback" earlier somewhere in the literature part. Without some mention of these concepts in advance, some readers may find it a bit not easy to follow your interpretation.
As the above conclusion "Chinese EFL learners.......... than learners from other cultures" involves comparison, it is better to provide some references to support this claim.
"Development-Approach was found only to positively predict Feedback Monitoring". This seems to show that development-approach did not positively predict direct feedback inquiry. It is better to interpret a bit why this was the case in your research context.
On p,12, you can elaborate on the pedagogical implications of the results.
Author Response
Response to Reviewer 1 Comments
Comments from Reviewer 1
Comments and Suggestions for Authors
This study investigated the relationships between feedback-seeking behavior, mindsets and achievement goals among Chinese EFL learners. As studies of these relationships have rarely been conducted in China, the study makes a contribution to the field. Specific comments are as follows:
Point 1: In the section of instruments, you may explain how you adapted the items of mindset and achievement goals, as well as items of FSB from Papi et (2019).
Response 1: We thank the reviewer for the suggestion. We have detailed the adaptation of the items of the three scales under Section 3.2, paragraph 2 from lines 256-265 on Page 6.
Point 2: Statistics about Mediating Role of Achievement Goals on Relationships Between Mindset and FSB were presented in Table 6. If possible, you may present these results in Figures which may help readers understand the results more easily.
Response 2: Thanks for the suggestion. We have added “Figure 1 Mediating Role of Achievement Goals on Relationships Between Mindset and FSB” on Page 10 as you suggested.
Point 3: In the Discussion part, on p.10, you wrote: "Chinese EFL learners who have been accustomed to examination-oriented and expository education (Liu, 2010) 391 may perceive higher ego, effort, and self-presentation cost of feedback than learners from other cultures". Perhaps it pays to mention concepts like "ego, effort, and self-presentation cost of feedback" earlier somewhere in the literature part. Without some mention of these concepts in advance, some readers may find it a bit not easy to follow your interpretation.
Response 3: We thank the reviewer for pointing out this issue. We have mentioned these concepts in the literature review of feedback-seeking behavior from lines 87-90 on Page 2. To help readers follow our interpretation, we added the definitions of these concepts following the sentence you mentioned.
Point 4: As the above conclusion "Chinese EFL learners.......... than learners from other cultures" involves comparison, it is better to provide some references to support this claim.
Response 4: Thanks for the suggestion. We have added two references as you suggested.
Point 5: "Development-Approach was found only to positively predict Feedback Monitoring". This seems to show that development-approach did not positively predict direct feedback inquiry. It is better to interpret a bit why this was the case in your research context.
Response 5: Thank you very much for this comment. We have added an interpretation of why development-approach did not positively predict direct feedback inquiry in our manuscript (Pages 10-11).
Point 6: On p,12, you can elaborate on the pedagogical implications of the results.
Response 6: Thanks for the suggestion. We have added section 6 entitled Educational Implications to our manuscript and elaborated on the pedagogical implications of our findings as you suggested (Page 12).
Reviewer 2 Report
Here are my comments:
#1 2.2 Instrumentation. The authors need to include criteria/references for determining high, moderate, and low reliability instead of saying it is high or moderate.
#2 2.2 Instrumentation. The authors might want to explain why two EFAs were needed for validating the instrument.
#3. 2.2 Instrumentation. The authors might also want to report the correlation of included items, at least the range of correlation. Normally, the researchers would reconsider the items lower than 0.3.
#4. In Table 1, the authors might want to add mode in the tables. The mode will be more meaningful than Min and Max.
#5 In Table 3. Sig, .000 should be reported as <.001, instead of .000. Same suggestion is applied to Tables 4-6.
#6 The author might want to reword "If our study could employ a mixed method combining self-reported questionnaire data with qualitative data such as interviews or learners’ diary, we might have richer findings." Also, some transitions might be needed.
Author Response
Response to Reviewer 2 Comments
Comments from Reviewer 2
Comments and Suggestions for Authors
Here are my comments:
Point 1: 2.2 Instrumentation. The authors need to include criteria/references for determining high, moderate, and low reliability instead of saying it is high or moderate.
Response 1: Thank you very much for this comment. We have included criteria for determining high, moderate, and low reliability as you suggested.
Point 2: 2.2 Instrumentation. The authors might want to explain why two EFAs were needed for validating the instrument.
Response 2: We thank the reviewer for pointing out this issue. We have explained the reason why two EFAs were needed in our manuscript as follows: we conducted two Exploratory Factor Analyses (EFA) for all three sub-scales for data reduction using SPSS 21 (IBM). In the first EFA, the method of Maximum Likelihood was used for extraction, direct oblimin with Kaiser Normalization for rotation, and eigenvalues larger than 1 (Kaiser’s criterion) and the scree plots were used to determine the number of factors. Results indicated that the number of factors presented by eigenvalues larger than 1 and the scree plots was contradictory. Based on the theoretical framework confirmed in previous studies (e.g., Papi et al., 2019, Korn & Elliot, 2016), we conducted the second EFA with a predetermined number of factors’ extraction.
Point 3: 2.2 Instrumentation. The authors might also want to report the correlation of included items, at least the range of correlation. Normally, the researchers would reconsider the items lower than 0.3.
Response 3: We very much appreciate your comments. We have reported the range of correlation of included items in Table 1. In the field of SLA, we commonly accept correlations between variables that are not very high because they might affect the process of foreign language learning, and we cannot ignore the important role of these variables even with not very high correlations (Qin & Bi, 2015).
Point 4: In Table 1, the authors might want to add mode in the tables. The mode will be more meaningful than Min and Max.
Response 4: Thanks for the suggestion. We have added a column of mode in Table 1.
Point 5: In Table 3. Sig, .000 should be reported as <.001, instead of .000. Same suggestion is applied to Tables 4-6.
Response 5: Thank you for the suggestion. We have revised it as you suggested.
Point 6: The author might want to reword "If our study could employ a mixed method combining self-reported questionnaire data with qualitative data such as interviews or learners’ diary, we might have richer findings." Also, some transitions might be needed.
Response 6: Thank you very much for identifying language-related issues. We have reworded the sentence and added transitions as you suggested.
Reviewer 3 Report
1. This paper mainly explores the relationships between achievement goals, mindset and feedback-seeking behavior or investigates the predictive role of achievement goals and mindsets on Chinese EFL learners’ FSB. A good tile usually can reflect the research content and focus. However, from your title, I am not clear what your study focus on. I suggest you make an amendment of the title.
2. In the paper, Section 1 Introduction is a big part, and most of them are the review of the literatures. I hope that the authors add one section “literature Review” and draft a small introduction.
3. Page 6. Line 266 ‘the paper reported the Cronbach’s alpha coefficients of mindsets scale is .52’. I think the Cronbach’s alpha coefficient (.52) is very low, even lower than the minimum acceptable value(0.6), representing the poor reliability. If a questionnaire has such poor reliability, I’m wondering how to make your findings convincing.
4. In Section 3 Results. The authors just present or list several tables and do not provide some basic and necessary analysis of data in the Tables. Due to lack of necessary explanations or interpretations of the data, readers cannot obtain general understanding of the features and distribution of the data.
5. The EFA Scree Plots are not necessary and can be removed from the Appendix.
Author Response
Response to Reviewer 3 Comments
Comments from Reviewer 3
Comments and Suggestions for Authors
Point 1: This paper mainly explores the relationships between achievement goals, mindset and feedback-seeking behavior or investigates the predictive role of achievement goals and mindsets on Chinese EFL learners’ FSB. A good title usually can reflect the research content and focus. However, from your title, I am not clear what your study focuses on. I suggest you make an amendment of the title.
Response 1: Thanks for the suggestion. We have revised the title as you suggested.
Point 2: In the paper, Section 1 Introduction is a big part, and most of them are the review of the literatures. I hope that the authors add one section “literature Review” and draft a small introduction.
Response 2: Thank you for pointing out this issue. We have drafted a small introduction to our paper and turned the original introduction into Literature Review section as you suggested.
Point 3: Page 6. Line 266 the paper reported the Cronbach’s alpha coefficients of mindsets scale is .52’. I think the Cronbach’s alpha coefficient (.52) is very low, even lower than the minimum acceptable value(0.6), representing the poor reliability. If a questionnaire has such poor reliability, I’m wondering how to make your findings convincing.
Response 3: Thank you very much for identifying this important issue. We have rechecked the 8 items included in the mindsets sub-scale. We found that four items related to a fixed mindset in the scale were negatively worded, which affected the reliability of the sub-scale. We have recoded the scores of the four items related to fixed mindsets and reanalyzed the reliability of the scale “Mindset” and found that Cronbach’s alpha coefficient was 0.81 rather than 0.52. We revised the data in our manuscript.
Point 4: In Section 3 Results. The authors just present or list several tables and do not provide some basic and necessary analysis of data in the Tables. Due to lack of necessary explanations or interpretations of the data, readers cannot obtain general understanding of the features and distribution of the data.
Response 4: Thank you very much for this comment. We have added some basic and necessary descriptions and analysis of data in the Tables to help readers gain a general understanding of the features and distribution of the data.
Point 5: The EFA Scree Plots are not necessary and can be removed from the Appendix.
Response 5: Thank you! We have removed the Scree Plots from the Appendix.
Reviewer 4 Report
Dear authors,
This manuscript was quite demanding in terms of conceptual framework, but I believe it should be published.
I only have 2 minor suggestions, which you might accept or ignore:
1. 296-297: it is not clear why researchers have “eliminated 13 random answers”.
2. 483-487: If your study could employ a mixed method combining self-reported questionnaire data with qualitative data such as interviews or learners’ diary, we might have richer findings. Future research could investigate the mediating roles of FSB on other variables such as motivation, language aptitude, proficiency levels, teacher-student relationship etc. as well as interventions that might improve L2 learners’ FSB in their language learning process.
Best wishes
Author Response
Response to Reviewer 4 Comments
Comments from Reviewer 4
Comments and Suggestions for Authors
Dear authors,
This manuscript was quite demanding in terms of conceptual framework, but I believe it should be published.
I only have 2 minor suggestions, which you might accept or ignore:
Point 1: 296-297: it is not clear why researchers have “eliminated 13 random answers”.
Response 1: We thank the reviewer for the positive comments. We have added the criterion of deleting random answers in our manuscript as follows: selecting the same answer for all the items.
Point 2: 483-487: If your study could employ a mixed method combining self-reported questionnaire data with qualitative data such as interviews or learners’ diary, we might have richer findings. Future research could investigate the mediating roles of FSB on other variables such as motivation, language aptitude, proficiency levels, teacher-student relationship etc. as well as interventions that might improve L2 learners’ FSB in their language learning process.
Best wishes
Response 2: We thank the reviewer for the suggestion and encouragement. We will further investigate the nature and role of FSBs that play in the process of foreign language learning.